# Bacterial bioindicators enable biological status classification along the continental Danube river

Laurent Fontaine[1], Lorenzo Pin[1,2], Domenico Savio[3,4,5], Nikolai Friberg[2,6,7], Alexander K. T. Kirschner[3,4,8], Andreas H. Farnleitner[3,4,5] & Alexander Eiler [1,9 ✉]

Despite the importance of bacteria in aquatic ecosystems and their predictable diversity patterns across space and time, biomonitoring tools for status assessment relying on these organisms are widely lacking. This is partly due to insufficient data and models to identify reliable microbial predictors. Here, we show metabarcoding in combination with multivariate statistics and machine learning allows to identify bacterial bioindicators for existing biological status classification systems. Bacterial beta-diversity dynamics follow environmental gradients and the observed associations highlight potential bioindicators for ecological outcomes. Spatio-temporal links spanning the microbial communities along the river allow accurate prediction of downstream biological status from upstream information. Network analysis on amplicon sequence veariants identify as good indicators genera *Fluviicola, Acinetobacter, Flavobacterium*, and *Rhodoluna*, and reveal informational redundancy among taxa, which coincides with taxonomic relatedness. The redundancy among bacterial bioindicators reveals mutually exclusive taxa, which allow accurate biological status modeling using as few as 2–3 amplicon sequence variants. As such our models show that using a few bacterial amplicon sequence variants from globally distributed genera allows for biological status assessment along river systems.

[1] Section for Aquatic Biology and Toxicology, Centre for Biogeochemistry in the Anthropocene, Department of Biosciences, University of Oslo, Blindernv. 31, 0371 Oslo, Norway. [2] Norsk Institutt for Vannforskning (NIVA) Gaustadalléen 21, 0349 Oslo, Norway. [3] Division Water Quality and Health, Department Pharmacology, Physiology and Microbiology, Karl Landsteiner University of Health Sciences, Krems an der Donau, Austria. [4] Interuniversity Cooperation Centre for Water and Health, Vienna, Austria. [5] Research Group for Microbiology and Molecular Diagnostics 166/5/3, Institute of Chemical, Environmental and Bioscience Engineering, TU Wien, Vienna, Austria. [6] Freshwater Biological Section, University of Copenhagen, Universitetsparken 4, Third Floor, 2100 Copenhagen, Denmark. [7] School of Geography, University of Leeds, Leeds LS2 9JT, UK. [8] Medical University Vienna, Institute for Hygiene and Applied Immunology, Water Microbiology, Kinderspitalgasse 15, 1090 Vienna, Austria. [9] eDNA Solutions AB, Kärrbogata 22, 44196 Alingsås, Sweden. ✉email: alexander.eiler@ibv.uio.no

In recent decades, the conservation and restoration of endangered river ecosystems have become pivotal around the globe. In Europe, the evaluation of ecological status relies on several indicators, including physical, chemical, and biological parameters. Biological indicators encompass different components of the communities living in freshwater ecosystems: fish, macroinvertebrates, aquatic flora including phytoplankton, phytobenthos and macrophytes. Little attention has been given so far to prokaryotes[1–3], with the exception of health-related water quality assessment, such as for bathing or drinking water[4–7].

Prokaryotes, due to their small size and high surface-to-volume ratio, are extremely sensitive to environmental changes, including variations in nutrients or pollutants even at very low concentrations[8]. This ability makes them perfect candidates as bioindicators and early warning sentinels, able to quickly respond to any sign of stress in the environment[9]. The need for biological indices based on prokaryotes has been stressed several times by different authors and multiple approaches have been proposed to introduce these communities in the current biomonitoring networks for freshwater as well as marine environments[10].

First attempts at identifying bacterial taxa suitable as bioindicators for biological status characterization were carried out by Fortunato et al.[11] in the Columbia River and its estuary. They were able to identify multiple taxa of bacterioplankton communities specific to various seasons and habitat types. Aylagas et al.[12] went a step further and used a biological status index based on benthic macroinvertebrates as a methodological basis to develop a comparable index using bacterial taxa from 16S rRNA amplicon sequencing. Bacterial taxa were divided into two ecological groups according to their positive or negative association with both organic and inorganic pollution inputs. The microgAMBI index they developed is based on the relative abundance of the taxa associated with each of the two ecological groups. In addition, this study revealed a significant correlation between the newly developed bacterial index and the traditional classification method based on macroinvertebrates, showing better performance of the former in some cases. This emphasized that bacteria and ecological status can be linked and that bacteria-based indices can be valid proxies for environmental impact assessment and water quality classification in coastal areas.

In a recent study, Cordier et al.[10] proposed combining environmental genomics and machine learning tools to develop reliable ecological quality status assessment routines. They suggested using supervised machine learning algorithms, which require a discrete variable (such as the ones for ecological quality status already in use) derived by continuous values of biological indicators obtained by morphological taxa identification. Decision tree learning avoids the black box problem associated with other algorithms as it allows to inspect individual trees of models and thus gain an understanding of the dynamics of relevant taxa. By assessing metabarcoding and morphological taxa identification at the same time, a predictive biomonitoring model could be built by training it on datasets with known ecological status classification. Firstly, such an approach could overcome several issues related to the lack of a taxonomic framework and the need for detailed taxonomic databases. These issues are solved by the model during the training phase since the ecological role of the taxa (for example operational taxonomic units or amplicon sequence variants) and their association with the whole community are automatically separated from the background noise[10]. Secondly, genomic data can be easily interpreted by managers without discrepancies typically associated with morpho-taxonomy. Therefore, data collection and processing could be fully automated. Thirdly, this molecular approach is also cost-effective and could be scaled up in time and space for consistent biomonitoring programs across countries and years.

In our work, we integrate these approaches using decision tree learning algorithms to detect potential bacterial bioindicators among the planktonic community of the Danube River from its source to the mouth. Here we seek to establish a method for identifying reliable bacterial bioindicators for the characterization and prediction of ecological patterns such as ecological status and water quality of the Danube River. We hypothesize the mobility of organisms within the river makes upstream community composition informative for downstream ecological outcomes such as ecological status from Saprobic index and chlorophyll *a* concentration. In addition, we applied multivariate statistics to detect suitable bioindicators based on the prevalence and variance of prokaryotic taxa[3] associated with environmental drivers, as a snap-shot approach not accounting for the spatial dependency of the sites. Finally, we compared the results from the snap-shot and spatio-temporal approaches to evaluate the efficiency and transferability of bacterial bioindicators for the assessment of river ecosystem's biological status as part of conservation and restoration efforts.

## Results

### Prevalence/variance analysis of the microbial community structure.
The results from the prevalence/variance analysis on amplicon sequence variant (ASV) relative abundance data revealed that the six most relevant phyla were, in this order, *Actinobacteriota*, *Bacteroidota*, *Verrucomicrobiota*, *Proteobacteria*, *Cyanobacteria*, and *Planctomycetota* (Fig. 1). *Actinobacteriota* for example significantly increased in relative abundance locally in the lower parts of the river. The dbRDA revealed a positive relationship between increased *Actinobacteriota* contribution and decreasing values of pH and water temperatures that characterize river reaches closer to the mouth. On the other hand, *Bacteroidota* had a higher contribution in the headwaters which appear to be correlated to higher loads of nitrogen, pH values and conductivity. Other phyla also showing high variances such as *Verrucomicrobiota* and *Proteobacteria* exhibited close associations to environmental parameters such as chlorophyll *a* concentration, high pH, and distance to river mouth, which itself was linked to increasing nutrient loads towards the upstream sites.

Accordingly, taxa with indicator potential for the assessment of the biological status could also be identified at lower taxonomic levels (i.e., class, family and genus level). At genus level (Fig. 2), the top six candidate bioindicators were: *hgcI_clade*, *CL500-29 marine group*, *Flavobacterium*, *Limnohabitans*, *Candidatus_-Methylopumilus*, and *Sediminibacterium*. These dbRDA results emphasize a differentiation of the bacterial community structure can be related to the succession of habitats along the river system using high (phylum) to low (genus) taxonomic resolution.

To investigate the non-linearity and suitability of bacterial beta-diversity for classification we performed modeling using extreme gradient boosting (XGboost). The model for pairwise bacterial community composition Bray–Curtis dissimilarities along the chlorophyll *a* gradient displayed clear non-linear patterns (Fig. 3) with an *R*-squared value of 0.32. We identified thresholds, where community composition changed more extensively than on average, at chlorophyll *a* concentrations of for example 1, 2.5, and 5.5 mg/l.

### Spatio-temporal identification of bacterial bioindicators.
The spatio-temporal approach identified several combinations of ASVs as good predictors for both water quality classification (according to the Saprobic Index) and chlorophyll *a* concentration. We focused the network analysis on the most informative ASVs as defined by a threshold of the 95th percentile in terms of

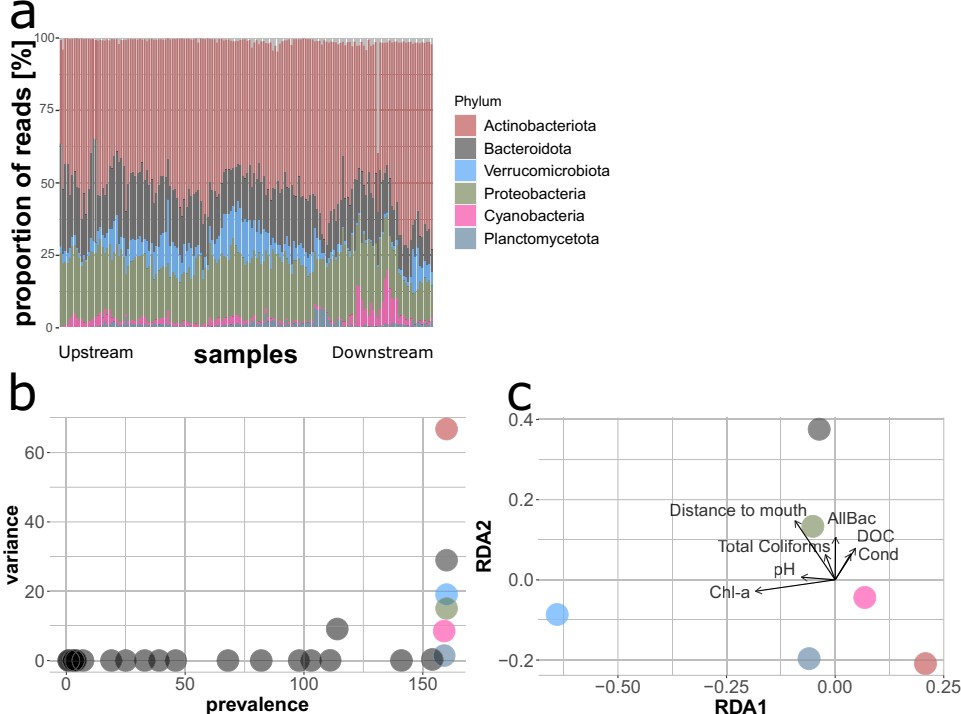

**Fig. 1 Visualization of the analysis on the prevalence and variance for most relevant bacterial phyla for the Danube River. a** Dynamics of the phyla among all the sampling sites. Samples are arranged from left (upstream) to right (downstream), with increasing distance from the source. **b** Bacterial phyla with the highest prevalence among the sampling sites and highest variance; the six colored dots are the taxa most suitable as biological indicators (highlighted in the ellipses) given their broad presence and wide variance across different environments. **c** The RDA shows the relationships of each of the 6 identified phyla with the environmental drivers. This provides the basis to link the dynamics of individual phyla with environmental properties; i.e., *Actinobacteria* contribute more to the community at the river mouth while *Bacteriodota* at the source and *Verrucomicrobia* increase in relative abundance with chlorophyll-a concentrations. Similar analyses on bacterial classes and families are presented in Supplementary Figs. 4 and 5.

occurrence in the predictive model outputs. As shown in Fig. 4, these ASVs formed a connected co-exclusion network. Although some of the ASVs seemed to be better predictors than others, none turned out to be indispensable to obtain the most accurate predictions. ASVs found to be top predictors could be assembled in combinations of two or three ASVs yielding the highest possible accuracy. The simplest predictive models yielding perfect accuracy were obtained with a *Fluviicola* representative paired with either an *Acinetobacter* or a *Flavobacterium*. *Rhodoluna* (class *Actinobacteriia*) and *Flavobacterium* (class *Bacteroidia*) were identified as the most important taxa, by frequency of occurrence, in terms of information content for biological status prediction. The other six of the 8 most informative ASVs belonged to *CL500-29 marine group* (class *Acidimicrobiia*), *Alterythrobacter* (class *Alphaproteobacteria*), *MWH−UniP1 aquatic group* (class *Gammaproteobacteria*) and two belonged to *Acinetobacter* (class *Gammaproteobacteria*).

A total of 168 ASVs in various combinations were found to yield perfectly accurate models for biological status classification, whereas 25 were present in the best model for chlorophyll *a*. The majority of the ASVs belonged to the phylum of *Proteobacteria*, with the classes of *Gammaproteobacteria* and *Alphaproteobacteria* being most abundant. Bacteria affiliated with class *Bacteroidia* were the second most relevant class by the number of ASVs identified as good predictors, followed by class *Actinobacteria*. As highlighted in Fig. 5, 20 ASVs affiliated with four bacterial classes yielded accurate predictions for both biological status classification and eutrophication, with most of them affiliated with the class of *Gammaproteobacteria*. One of these, an *Acinetobacter* representative, is also listed among the top predictors suggested

by the network analysis. Class *Bacteroidia* was the second most important class in terms of the number of ASVs being good predictors for both biological status classification and chlorophyll *a* concentration, followed by *Verrucomicrobiae* and only one ASV belonging to class *Bacilli*. Five ASVs belonging to the *Actinobacteria* were identified as good predictors only for chlorophyll *a* concentration.

The predictive accuracy of the models was unequal across the different transects of the river for chlorophyll *a*. While the models for only the center transect yielded an *R* squared of 0.98, the global *R* squared for all three transects was 0.86. In contrast, the accuracy of the best predictive models for water quality classification was always 100% for the three transects (left, right and center observations).

**Discussion**

This study highlights the potential of the identification of bacterial bioindicators for assessing the biological status of river ecosystems from bacterial metabarcoding data, despite methodological constraints such as copy number variations[13], PCR biases[14], the compositional character of the sequencing data[15] as well as other limitations[16]. Both the prevalence/variance and spatio-temporal approaches used in this study yielded comparable results in terms of candidate bacterial bioindicators, showing that dominant bacterial taxa can be ecologically informative. As such both approaches could complement and expand the currently implemented morphology-based methods to produce ecological quality assessment, but in a faster and more cost-effective way. The spatio-temporal approach is a fully data-driven approach classification system and can be iteratively updated,

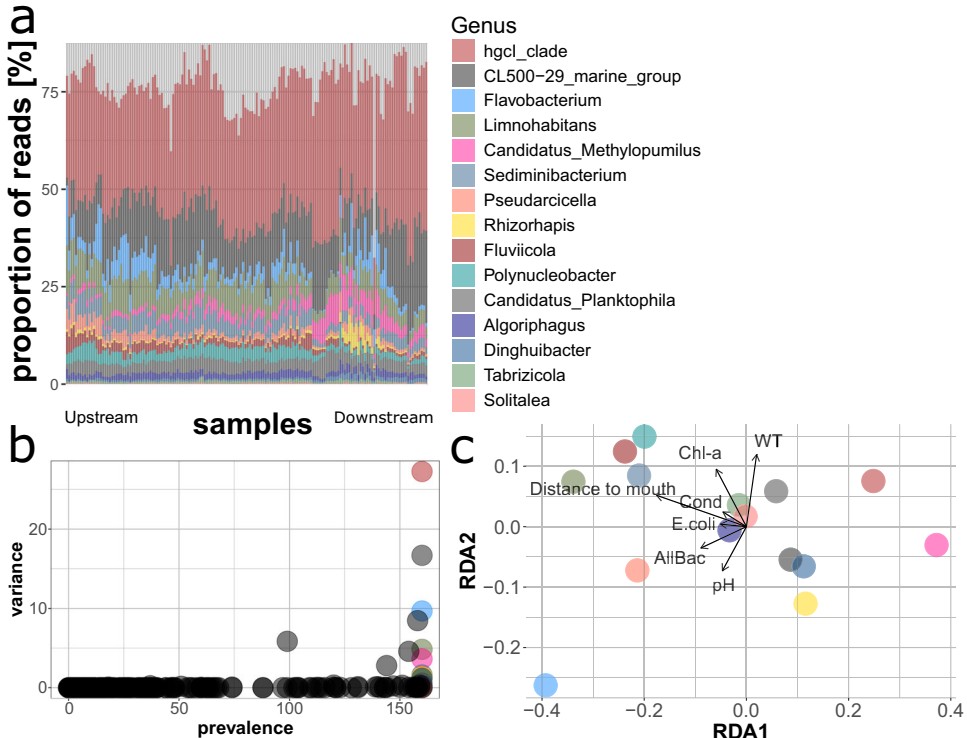

**Fig. 2 Visualization of the analysis on the prevalence and variance for the most relevant bacterial genera for the Danube River. a** Dynamics of the genera among all the sampling sites. Samples are arranged from left (upstream) to right (downstream), with increasing distance from the source. **b** Bacterial genera with highest prevalence among the sampling sites and highest variance; the 15 colored dots are the taxa most suitable as biological indicators (highlighted in the ellipses) given their broad presence and wide variance across different environments. **c** The RDA shows the relationships of each of the 15 identified genera with the environmental drivers. This provides the basis to link the dynamics of individual genera with environmental properties; i.e., *Canditatus Methylopumilus* contribute more to the community at the river mouth while *Limnohabitans* at the source and *Flavobacterium* spp. increase in relative abundance with pH. Similar analyses on bacterial classes and families are presented in Supplementary Figs. 4 and 5.

which can unlock the disruptive potential of metabarcoding and machine learning for biomonitoring. Samples from routine monitoring can in real-time complement the training data and further improve model predictions, and in the long run cover most of the possible environmental condition space.

Potential future improvements to our approach include human interpretability, and facilitating understanding of the trained models. This is crucial for a smooth transition to machine learning approaches for complex decision-making problems[17]. There are now multiple tools available that allow to explain the predictions, by identifying the features of the data that contribute to a given prediction[18]. Such approaches will further allow biologists and ecologists to evaluate and control model development, without falling into the caveats of machine learning[10].

By using the *spatio-temporal* approach, accurate modeling of biological status and chlorophyll *a* concentration can be achieved based on only small sets of ASVs, provided they have high predictive power and are complementary in terms of information content. That accurate biological status modeling can rely on as few as 2–3 ASVs opens opportunities for various approaches to bioindicator identification. These include on-site identification systems using biosensors such as ecogenomic sensors or small microfluidic lab-on-chip devices that could target identified bioindicators for general ecological status classification. Point of care and real-time sensing devices could unlock the real potential of microbes in current biomonitoring networks and biodiversity forecasting as already seen in weather forecasting.

The prevalence/variance analysis at class and genus level revealed several taxa affiliated with the dominant phylum of *Actinobacteriota* to be the most informative for changes in environmental properties. The *Actinobacteriota* phylum embraces taxa typically occurring in several freshwater environments[19] and holds a central role in the heterotrophic biogeochemical processes within river ecosystems[20]. They seem to be strongly influenced by pH and nutrient concentrations, which often negatively affect their abundances due to their low growth rates[19,21–23] and the results from our analysis confirm their environmental preferences. Their higher abundance in the lower reaches of the Danube River may be related to the lower nutrient concentrations and organic matter content as a result of dilution effects, while their lower abundances in the upstream river reaches might originate from the negative association between suspended particles and this taxon[11,24,25].

When focusing on the axes of the RDA visualization (Figs. 1 and 2), *Bacteroidota* were opposing this trend and were more related to the upper river reaches with higher nutrient inputs, conductivity, and strong dependence on nearby forests and groundwater sources[26]. *Flavobacterium* (class *Bacteroidia*) was the third important genus to be highlighted from the prevalence/variance analysis with the highest abundances in the upstream river reaches, thus showing a preference for more eutrophic environments. The abundance pattern for this phylum is likely related to the prevalent inputs of recalcitrant organic matter from the riparian zone and forests. From the literature, the headwater microbial community likely has its main source in the soil and groundwater communities drifting into the upper reaches of the Danube River[26,27]. *Bacteroidota* already previously have been shown to be good indicators for anthropogenic pollution and impacts from agriculture on freshwater ecosystems[28]. Real-time qPCR is usually used to identify gene markers related to this

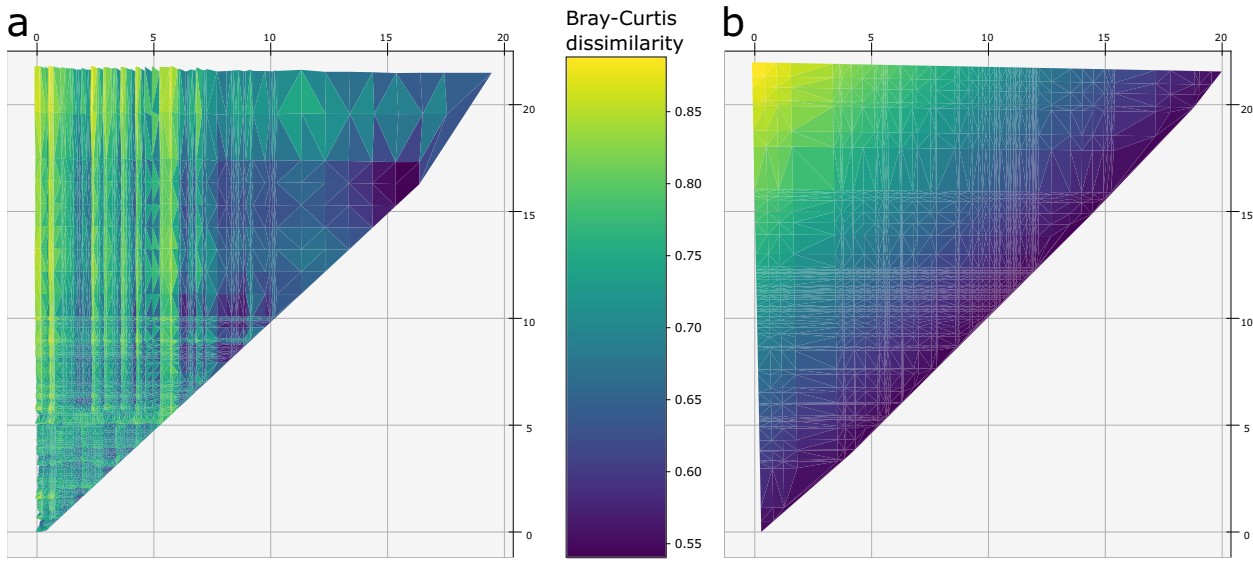

**Fig. 3 Threshold analysis for microbial beta diversity (Bray–Curtis dissimilarities) along the chlorophyll *a* gradient.** The color scale represents Bray–Curtis dissimilarity values computed between sites. The axes stand for chlorophyll *a* concentration (mg L$^{-1}$) at each site. In (**a**), the observed beta diversity patterns are presented along the chlorophyll *a* gradient, as modeled using XGboost. In (**b**), a hypothetical relationship is represented where dissimilarity between communities increases linearly as a function of the difference in chlorophyll *a* between sites. The mean value of the response surface in (**a**) can be treated as the baseline beta-diversity across all sites. Data points with values below the mean represent higher similarity between sites; likewise, higher values represent lower similarity. To interpret the response surfaces of observed values, one may begin by looking at a point bordering the diagonal and then follow a line of points further up on the chlorophyll *a* axis. Here, a ridge indicates a chlorophyll *a* concentration to be a likely threshold from which the shift in bacterial community composition is greater than average. In the same manner, a valley indicates a chlorophyll *a* concentration likely located on an interval of the chlorophyll-a gradient along which bacterial communities do not shift substantially. More details on the XGboost approach and its interpretation are given in Fontaine et al.[57].

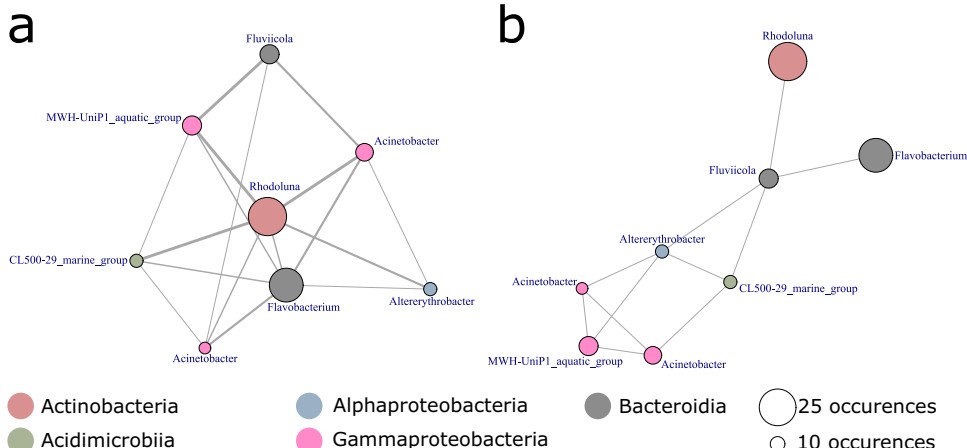

**Fig. 4 Network analysis results for the most frequent ASVs returned by the spatio-temporal approach yielding perfectly accurate water quality classification.** Colors represent the respective bacterial classes to which the ASVs belong. **a** Co-occurrence network, node diameter represents the number of occurrences of each taxon in the best models, while the thickness of the connecting lines represent the number of co-occurrences of ASV pairs. **b** Co-exclusion network. Links represent taxa that never occurred together in a same model.

taxon within river systems impacted by wastewater treatment plant effluents and in aquaculture systems, showing *Bacteroidota* to be reliable in determining the degree of pollution[29–33]. In our study, taxa belonging to the class *Bacteroidota* were clearly related to the biological status of the Danube, further confirming their potential relevance for biomonitoring in river ecosystems.

*Proteobacteria* were another group highlighted as bioindicators by the variance/prevalence analysis, with the genus *Limnohabitans*—a representative of the class *Gammaproteobacteria*—

identified as an indicator for eutrophic conditions. Other members of the class of *Gammaproteobacteria*, such as *E. coli* are important and sensitive indicators for fecal pollution in aquatic ecosystems and water resources. Their numbers are traditionally surveyed by standardized culture-based enumeration methods[5,34]. Very interestingly, *Cyanobacteria* had relatively low abundances in river sections characterized by higher chlorophyll *a* concentration. This pattern may seem counterintuitive but is most likely related to green algae and diatom abundances in the

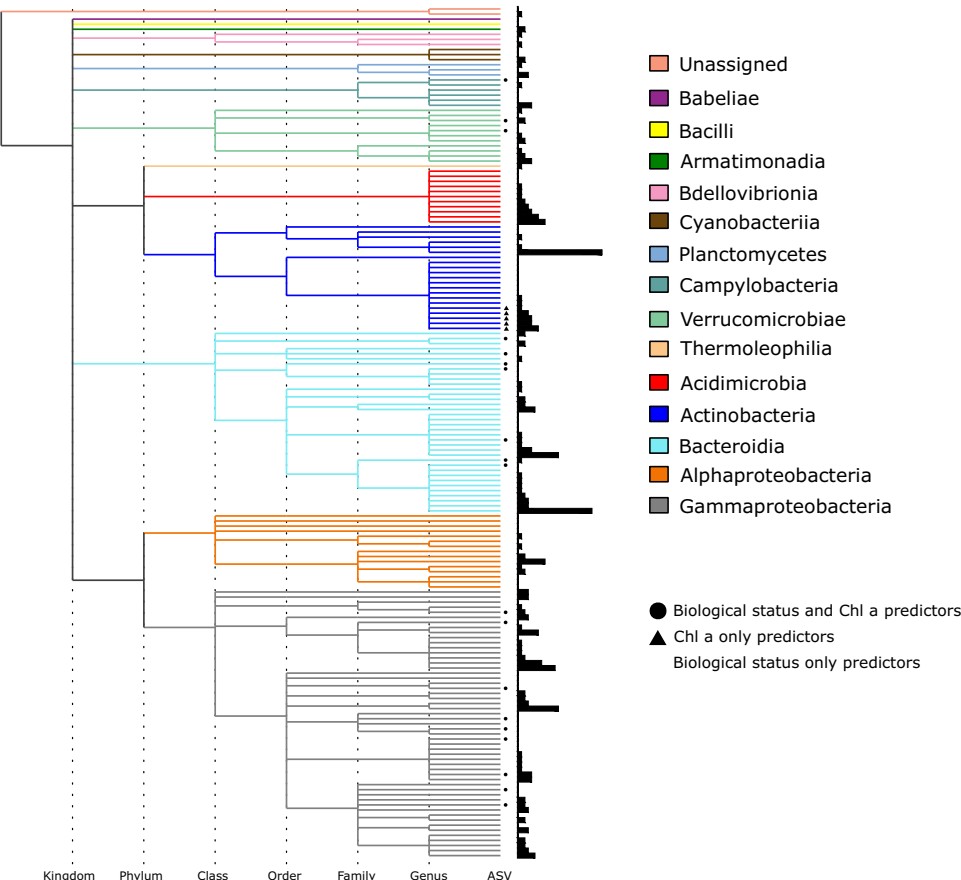

**Fig. 5 Phylogenetic analysis of the bioindicator taxa identified for the best predictive models for biological status classification and eutrophication (chlorophyll _a_ concentration).** Colors represent bacterial classes while symbols represent the variable for which an ASV was identified as a predictor. The horizontal position of nodes in the dendrogram indicates at which taxonomic level they occur, marked with dotted lines from Kingdom to genus. The histogram represents the relative frequency of each ASV's occurrence in model outputs.

phytoplankton community, which had a greater impact on the chlorophyll _a_ concentrations compared to their bacterial counterparts[35].

From both approaches implemented in this study, it seems that ASVs belonging to the dominant taxa in the bacterial community such as _Actinobacteriota_, with the genus _Rhodoluna_ in particular, _Bacteroidota_ with particularly the genus _Flavobacterium_ as well as _Proteobacteria_ are also among the best indicators for shifts in the biological status of water quality in the Danube River. This finding aligns with the results from Fortunato et al.[11], where among the microbial communities from three different environments embracing a salinity gradient, the best indicators for changes in the environmental drivers were among the most abundant taxa.

Although the spatio-temporal approach identified multiple combinations of ASVs yielding accurate predictions, many frequently appearing ASVs never occurred together in any given model. It can be assumed these ASVs do not co-occur in models because they hold redundant information content. The co-exclusion network (Fig. 4) allows for the identification of redundancies in information content among the most informative ASVs. These results from the network analyses suggest that there are no truly best nor indispensable bacterial taxa yielding information on the biological status and chlorophyll _a_ concentration. Rather, there is a certain amount of information required to get accurate models which is shared across multiple taxa with various degrees of overlap.

While functional information on bacterial communities was not available in this study, it is conceivable that ecological

function and information content are linked. Redundant ASVs ought to be taxa occupying similar ecological niches albeit without experiencing competitive exclusion. Redundancy, and as such overlap in ecological information content, is likely to increase with phylogenetic relatedness among taxa, as visualized by a larger number of connections among closely related taxa in the co-exclusion network. Among the best predictors, representatives of same classes showed higher redundancy relationships compared to taxa belonging to different phyla (Fig. 5). This has been shown in a previous study where functional coherence among closely related taxa ranged from low (species) to high (phylum) taxonomic levels[36]. At the same time, bacteria can belong to different ecological groups while being taxonomically distinct at the level of a single nucleotide in a 16S rRNA tag[37].

Functional coherence can be obscured within taxonomic datasets by gene degeneracy and horizontal gene transfer. Yet, studies comparing bioassessment performances of different taxonomic levels showed that some higher taxa (in particular genus level) might be relatively precise and efficient compared to species level in the eukaryotes[38,39]. Others emphasized the advantage of mixed taxonomic levels to adapt the taxonomic detail to the information content for each clade[40]. Redundant ASVs belonging to the same high taxonomic level as identified in our study confirm the potential for high information content at high taxonomic levels. Furthermore, these taxonomic groups may contain bioindicators of high information content across a wide range of river systems even though the exact same ASVs as identified in our study are not present.

In order to identify the true nature of the signal behind informational redundancy between taxa, it can be argued that the entire genomic information must be obtained of microbial communities. This ideal case of having access to complete genomic information would allow for identifying shared functional traits and more generally partitioning the entire information content into discrete components. While the latter is currently unrealistic, one can begin the search for ideal bioindicators with a hypothesis that clusters of orthologous groups of genes ought to be superior to 16S rRNA ASVs. Yet, functional genes on their own may not necessarily turn out to be a unit of ecological information as they can be degenerate or part of incomplete metabolic pathways.

The community dynamics shown in our study confirm the findings of previous bacterioplankton surveys in other river ecosystems[41–43] as well as from the Danube River itself[26]. It seems that even across different river ecosystems around the globe, the bacterioplankton succession follows similar trajectories, with upstream and downstream communities being quite different and strictly related to the water residence time[43]. As hypothesized by Read et al.[43], upstream river reaches could be characterized by fast growing bacterial taxa, r-strategists capable of processing the resources; these would be the *Bacteroidota*. While on the other hand, when the river becomes larger, the influence from the riparian zone is lower and nutrient availability is reduced due to dilution effect and reduced bioavailability of recalcitrant carbon sources; k-strategist organisms such as the *Actinobacteriota* would prevail. This recurring pattern of microbial succession suggests spatio-temporal predictive modeling for all types of biological and ecological status assessment is applicable to river ecosystems sharing a similar bacterioplankton composition.

Microbial community assembly across gradients may follow relationship patterns ranging from broadly linear to clear-cut non-linear. It thus raises the question of the nature of such relationships in this study and how to model them in a way that is coherent with natural patterns. When relationships are of linear type, one should model a continuous response variable as it avoids unnecessary loss of information from discretizing a gradient into classes. Likewise, pronounced non-linearity is fertile ground for a classification approach.

In our study, the occurrence of abrupt and below-average rates of change in microbial community composition along the chlorophyll *a* gradient indicates bacterial communities tend to be somewhat homogeneous along certain river sections along the flow path and change markedly beyond specific thresholds. For example, the last clear shift occurs between 5 and 6 mg of chlorophyll *a* per liter, after which the bacterial community rate of change remains roughly below average until the end of the gradient (Fig. 3). Concentrations of 6 mg chlorophyll *a*/l and above can be construed as the natural range corresponding to eutrophic waters. The non-linear relationship between bacterioplankton community composition along the chlorophyll *a* gradient supports a general biological classification approach albeit leaving unanswered how many natural classes the latter possesses and if the reference Saprobic Index classes overlap natural ones.

The threshold analysis treated observations as independent, whereas they are linked in space and time, meaning discrepancies in community assembly reaction time between primary producers and the rest of the microbial community may occur as water masses travel and conditions change faster than equilibrium can be reached. In cases of merging tributaries, water masses may initially flow side-by-side and mix thoroughly only after many kilometers[44]. Our results from the spatio-temporal approach regarding chlorophyll *a* point in this direction. This laminar vs mixing duality is reflected in higher model accuracy for the midstream transect and must be considered when creating design matrices. Left, right and midstream observations should thus be treated as separate spatial series in terms of observation order. Yet, overall accuracy is greater when integrating distinct time series into one model rather than processing each separately. To achieve accuracies for the left and right transects that are comparable to the midstream, it is conceivable that the values used for lag steps and scaling at tributary merging locations may have to come from upstream sites of the tributary itself rather than the main river.

To verify if bacterial indicators are robust for biological status prediction, one would need to test whether the clades underlying given ecological processes in a river system apply in the same fashion to other rivers and across seasons. To uncover the suitability of candidate ASVs highlighted in this study as well as the general applicability of variance-based and spatio-temporal ASV screening methods, extensive training data is required from multiple river systems in order to develop general and robust classification systems for environmental assessment based on bacterial data. There is also a need for harmonization of sampling protocols, DNA library preparation, sequence data generation, and downstream bioinformatics processing to achieve reproducible biological status assessment across river systems.

## Methods

**Study area**. The Danube River spans 2780 km, from its source in the Black Forest in Germany from whence it flows across ten different countries (Germany, Austria, Slovakia, Hungary, Croatia, Serbia, Bulgaria, Romania, Moldova and Ukraine), ending its course in a delta on the Black Sea. Its catchment area covers 801,500 km$^2$ and is populated by almost 81 million inhabitants[45]. The water from the Danube River serves industries, agriculture and people as drinking water supply and recreational area, as well as a transportation route connecting several countries[30].

Every six years since 2001, the Joint Danube Survey (JDS) has been organized by the International Commission for the Protection of the Danube River (ICPDR)[30,45,46]. As an extensive survey of 2600 km of the Danube River (Supplementary Fig. 1), the JDS is one of the largest international scientific expeditions, aiming to collect data on the river hydro-morphology, dynamics of several biological communities, water physico-chemical parameters, pollutants, and more.

Water samples for the microbial community were collected by hand from the epilimnion in 1 L flasks, at the same time as the physico-chemical parameters of the water were also measured with hand probes. Macroinvertebrates samplings were performed using a Multi Habitat Sampling approach[47], where different parts of the riverbed were disturbed, and the macroinvertebrates were collected with a net. Left and right sides of the river, together with its center, were sampled at 60 locations.

All data, sampling methods, as well as analytical methods, are publicly available via the official website of the International Commission for the Protection of the Danube River (ICPDR; http://www.icpdr.org/wq-db/) and the final scientific report[48]. Selected data from JDS3 (2013) were published previously in several studies[30,35].

Ecological status classification compliant with the Water Framework Directive was performed from JDS3 data collected for some of the biological communities analyzed along the river, among which macroinvertebrates, macrophytes, fish, and phytoplankton. Different biological classification systems are used by different countries and intercalibrating the different methods remains challenging[48]. The Saprobic Index, based on benthic

macroinvertebrate communities, is one of the best-established classification systems to assess biological status in compliance with the WFD and is mainly used to assess organic pollution. This was the only classification system used consistently throughout the various countries along the Danube River and was therefore used as the reference biological classification system in our study.

**DNA extraction and 16S rRNA gene amplicon library preparation.** Genomic DNA was extracted using a slightly modified protocol of a previously published phenol-chloroform and bead beating-based procedure[49] using isopropanol instead of polyethylene glycol for DNA precipitation[50]. Total DNA concentration was assessed applying the Quant-iT™ PicoGreen® dsDNA Assay Kit (Life Technologies Corporation, USA) and 16S rRNA gene concentrations in the DNA extracts were quantified using domain-specific quantitative PCR where reactions contained 2.5 μL of 1:4 and 1:16 diluted DNA extract as the template, 0.2 μM of primers 8F and 338 targeting the V1–V2 region of most bacterial 16S rRNA genes and iQ SYBR Green Supermix (Bio-Rad Laboratories, Hercules, USA)[26]. DNA extracts were normalized with regard to 16S rRNA gene concentrations in order to use standardized numbers of bacterial 16S rRNA gene templates for amplification and barcoding in a two-step barcoding procedure. In short, the first-step primers (341f and 805r) contained adapters for introducing Illumina adapters and dual barcodes were used in the second step. The first-step PCR primers were thus (adapter sequence, followed by primer sequence) adapter-341f ('5 - ACA CTCTTTCCCTACACGACGCTCTTCCGATCTNNNNCCTAC GGGNGGCWGCAG-3′) and adapter-805r ('5-AGACGTGTGCT CTTCCGATCTGACTACHVGGGTATCTAATCC-3′). The first step amplicon PCR (ampPCR) was carried out in duplicate in 20 μL reaction mixtures containing 1 × Q5 reaction buffer, 0.2 mM dinucleoside triphosphates (dNTPs), 0.5 μmol L$^{-1}$ forward and reverse primers, 0.4 U of Q5 high-fidelity DNA polymerase (New England BioLabs), as well as environmental DNA as template that was normalized to equal amounts of 16S rRNA gene copies prior to barcoding in order to increase comparability and reduce PCR bias. Cycling conditions for 1st-step ampPCR were 98 °C for 1 min, followed by 20 cycles of 98 °C for 10 s, 62 °C for 30 s, 72 °C for 30 s, and a final extension at 72 °C for 2 min. The duplicate products were pooled and purified using the Agencourt AMPure XP purification system (Beckman Coulter). The second PCR step containing variable combinations of primers with different multiplex-identifiers for sample-specific 'barcoding' (forward, AATGATACGGCGACCACCGAGATCTACAC-[index]-A CACTCTTTCCCTACACGACG; reverse, CAAGCAGAAGAC GGCATACGAGAT-[index]-GTGACTGGAGTTCAGACGTGT GCTCTTCCGATCT) binding to the first-step adapters and incorporating Illumina adapters was carried out in single 20 μL reactions. Reactions contained 1 × Q5 reaction buffer, 0.2 mmol L$^{-1}$ dinucleoside triphosphates (dNTPs), 0.25 μmol L$^{-1}$ forward and reverse index primers, 0.4 U of Q5 high-fidelity DNA polymerase (New England BioLabs) and 2 μL of purified amplicons from 1st-step ampPCR as template. Cycling conditions for the 2nd-step index PCR (idxPCR) were 98 °C for 1 min, followed by 15 cycles of 98 °C for 10 s, 66 °C for 30 s, 72 °C for 30 s, and a final extension at 72 °C for 2 min. PCR products (amplicon libraries) were purified as described above and quantified with the PicoGreen kit (Life Technologies). Products were sequenced at the SciLifeLab SNP/SEQ sequencing facility at Uppsala University, Uppsala, Sweden, on an Illumina MiSeq (2 × 300 bp) in two runs.

**16S rRNA gene amplicon data analysis.** Raw sequence data from JDS3 from different sequencing runs were processed separately. After automatic demultiplexing of raw amplicon sequencing data

by the Illumina MiSeq sequencing software into 675 individual samples (including 66 technical and 329 biological replicate pairs) from two sequencing runs for JDS3, primers were removed using the CUTADAPT tool[51] and sequences without matching primers were discarded. The R package dada2 (version 1.8)[52] was used for de-replication, denoising and sequence pair concatenation. After manual inspection of quality score plots, forward and reverse reads of the bacterial 16S rRNA gene amplicons were trimmed at 260 and 200 bp length, respectively, and reads with a single phred score below 10 were removed. After de-replication of reads, forward and reverse error models were created using a subset of ~10$^7$ sequence reads. Chimeras were removed using the "removeBimeraDenovo" function in "dada2." Taxonomy was assigned using the Bayesian classifier and SILVA non-redundant database 138[53,54]. Next, 329 biological replicate pairs for the JDS3 dataset were merged and averaged to obtain the final Amplicon Sequence Variants (ASVs) table. The dataset considered for this study comprised the left, right and center transects for the Danube River, amounting to 160 samples in total, after exclusion of samples from tributary sites. Chloroplast, mitochondrial, eukaryotic as well as archaeal ASVs were removed using the *phyloseq* package version 1.40.0 resulting in 3509 bacterial ASVs. Details on sample and sequence data are given in Supplementary Table 1.

The code used for sequence data processing is available on github—https://github.com/alper1976/danube_indicators (https://doi.org/10.5281/zenodo.8193431)[55].

**Statistics and reproducibility.** Statistical analyses and plot generation were conducted in R version 4.0.2 (2019-12-12; R Core Team, 2014). From the ASV table, we created four datasets, clustering the ASVs on phylum, class, family and genus levels. For each of the four datasets we calculated the prevalence (occurrence frequency at each sampling site) and coefficient of variation (standard deviation of taxon abundance divided by the mean)[3]. These two parameters were plotted against each other in a scatter plot to detect taxa with the highest variance occurring more often across all sampling sites, as these might represent suitable biological indicators.

To identify the environmental drivers behind bacterial community assembly, we selected specific variables among the metadata ("pH," "Electric Conductivity," "water temperature," "River km/Distance to mouth," "Ntot," "Chl a," "DOC," "Total Coliforms (LOG10(x + 1))," "BacHum (LOG10(x + 1))," "AllBac (LOG10(x + 1))," "E.coli (LOG10(x + 1))") (Supplementary Fig. 2). Missing observations were imputed with the *mice* function from the 'namesake' R package (version 3.13.0) and the VIM package (version 6.2.2), using the Predictive Mean Matching imputation approach[56]. Relationships between beta diversity and chlorophyll *a* were investigated in order to characterize the extent of linearity along this gradient and look for possible thresholds associated with shifts in microbial community composition. The analysis was performed using pairwise bacterial community composition Bray-Curtis dissimilarities along the chlorophyll *a* gradient according to Fontaine et al.[57].

**Prevalence/variance analysis of the microbial community structure.** By using a step forward selection model (*vegan* version 2.6-2), we selected the variables most related to the Hellinger-transformed Bray-Curtis dissimilarity matrix for each of the four bacterial taxonomic levels. The aim of forward selection was to identify environmental parameters covarying with community composition. Next, the most informative variables were tested for multicollinearity by using the variance inflation factor value and tolerance value. Those variables showing a variance inflation factor value above 5 were excluded from further

analyses. A distance-based Redundancy Analysis (dbRDA) was performed by using the Hellinger-transformed Bray–Curtis matrix for each of the four taxonomic levels by only including the most prevalent and variable taxa with the above selected metadata to identify specific relationships amongst phyla, classes, families or genera with particular environmental drivers.

**Spatio-temporal identification of bacterial bioindicators.** Biological status-related variables (chlorophyll *a* concentration as a proxy for eutrophication and classification of saprobity according to the Saprobic Index) for downstream sites were predicted using information from upstream bacterial community composition. The identification of ASVs informative for biological status prediction was performed in two main steps. At first, we created an unsupervised random forest model (*randomForest* version 4.6–14) in order to gain knowledge on the latent structure of the sites based on ASVs, in order to prefilter the ASV table due to the following screening step being computationally demanding. Without knowledge of the true structure within the data, a grid search optimization of parameters for the unsupervised random forest model was performed on a supervised model where the response variable was chlorophyll *a* concentration, and the explanatory data was the Hellinger-transformed ASV abundance table. The lowest mean squared error of these supervised learning models was obtained with a combination of mtry=1600 and ntree=30 with a 0.8/0.2 dataset split for training and testing sets. Using these parameters, the unsupervised random forest model was trained on the same Hellinger-transformed ASV abundance table. From the unsupervised model output, candidate ASVs for downstream analysis were selected based on exceeding a threshold of 0 in percentage increase of mean squared error (%IncMSE).

The second main step in the ASV screening process was to run, for each ASV in decreasing order of %IncMSE, a grid search for the optimal combination of steps (i.e., number of upstream sites) to use for (i) transformation and (ii) lag using XGboost (sci-kit learn implementation of XGboost version 1.3.3). In other words, the objective of the grid search is to find the two optimal shifting frames of number of sites for each ASV used for predicting response variables. The transformation in this case is scaling and centering of ASV abundances, while lag is the number of observations needed from upstream sites for the prediction at a given downstream site. The criterion for selecting the best combinations of values for transformation and lag was the model *R* squared on the test set for chlorophyll *a* and accuracy percentage for biological status classification. The design matrix for the ASV screening initially contains sampling ID, transect code (left, right, and center), and distance to the river mouth. These metadata variables are included to account for the continuity of each longitudinal transect in terms of laminar flow dynamics as well as travel time in the case of distance to the river mouth. Upon yielding an improved model *R* squared, an ASV is added to the design matrix in its optimal transformation and lag combination. The ASV screening process was run 1000 times, for both water quality variables, with random ASV order and random combinations of XGboost hyperparameters. However, for the classification based on the Saprobic Index, two observations presented high biological status and first appeared at 550 km from the river mouth (Supplementary Fig. 3), which is past the first 80 % of observations, by order of distance from the Danube source. It is impossible to evaluate model accuracy if the test set contains variable classes which are not present in the training set. Thus, in order to include observations of all variable classes in the training set, the training and testing split of the data for XGboost models (both for chlorophyll *a* concentration and biological

status classification) was set at 0.85 and 0.15 respectively, with the split occurring after the first 85% of observations.

The model outputs for the ASV screening for biological classification were filtered to keep only those presenting an accuracy of 100 %. In the case of chlorophyll *a*, the model yielding the highest R squared was kept. The model outputs were turned into a presence/absence table with individual models as rows and ASVs as columns for water quality classification. The resulting table was then used for a network analysis, performed using R library igraph (version 1.2.4.2). Co-occurrence and co-exclusion networks were used to investigate information content and redundancy between ASVs. Informational redundancy between individual taxa, in the sense of information content translating into model predictive power, was here interpreted as ASVs which are never present together in any individual screening process output. The logic behind this assumption is that ASVs are mutually exclusive if they contain shared predictive information. As the screening process will pick up the first ASV of a set containing a given share of information, the following ones from that same set would be left out since they would not improve the model.

In opposition to co-occurrence where there is certainty about said relationship, the same cannot be assumed from co-exclusion found here since it could result from the limited number of random permutations of ASVs screened together. Nevertheless, a small number of ASVs appeared in most of the screening process outputs, suggesting a semblance of saturation in their sampling and thus that their co-exclusion relationships are accurate. The selection of ASVs kept for this co-exclusion analysis was thus set at the 95th percentile by number of occurrences. A dendrogram was then built with all the ASVs yielding the best models for eutrophication and water quality to visualize taxonomic overlap between predictors for both variables. The hierarchical clustering was performed on a distance matrix computed from the taxonomy table.

**Reporting summary.** Further information on research design is available in the Nature Portfolio Reporting Summary linked to this article.

## Data availability

Data from the ICPDR that were used in this manuscript can be found on GitHub—https://github.com/alper1976/danube_indicators (https://doi.org/10.5072/zenodo.1222217)[55]. All sequencing data are available in NCBI Sequence Read Archive under accession number PRJNA835446. Additional data can be obtained upon request from the ICPDR. The source data behind histograms in Figs. 1, 2, and 5 are available in Supplementary Data 1–3, respectively.

## Code availability

Code used for sequencing data processing, modeling, statistical analysis and additional detail on the approaches are available on GitHub—https://github.com/alper1976/danube_indicators (https://doi.org/10.5281/zenodo.8193431)[55].

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

## Acknowledgements

This study has been supported financially by the Department of Biosciences and the Centre of Biogeochemistry in the Anthropocene, University of Oslo ("Startpakke" of Alexander Eiler). This research is part of the EuroFLOW project (EUROpean training and research network for environmental FLOW management in river basins) funded by the European Union's Horizon 2020 - Research and Innovation Framework Programme under the Marie Skłodowska-Curie grant agreement (MSCA) No.765553. This study was supported by the Austrian Science Fund (FWF) as part of the "Vienna Doctoral Program on Water Resource Systems" (DKplus W1219-N22) and the FWF projects P25817-B22, P23900-B22 and P32464-B, as well as the research project "Groundwater Resource Systems Vienna" in cooperation with Vienna Water (MA31). Infrastructure (cruise ships, floating laboratory) and logistics for collecting, storing and transporting samples were provided by the International Commission for the Protection of the Danube River (ICPDR). This manuscript includes data licensed by the ICPDR, retrieved from the Danubis database (https://danubis.icpdr.org/). We specifically thank Wolfram Graf, Béla Csány, Patrick Leitner, Momir Paunovic, Thomas Huber, Joszef Szekeres, Claudia Nagy and Peter Borza for providing macroinvertebrate data and saprobic index classification, Martin Dokulil and Ulrich Donabaum for providing chlorophyll *a* data, Carmen Hamchivici, Florentina Dumitrache, Fabio Sena, Günther Umlauf, Carmen Postolache and Ion Udrea for providing general physico-chemical parameters and nutrients as well as Marija Marjanovic-Rajcic and Damir Thomas for providing DOC data. We thank Franck Lejzerowicz for insightful suggestions which improved the manuscript.

## Author contributions

D.S., A.K.T.K., and A.H.F. designed the survey as well as managed and performed sampling. D.S. performed the lab work for processing DNA samples. L.P., D.S., and A.E. processed the sequencing data. L.F. ran the spatio-temporal modeling and threshold analysis. L.P. ran the prevalence/variance analysis. A.E., N.F., A.K.T.K., and A.H.F. obtained funding for the survey. L.F., L.P., D.S., and A.E. wrote the manuscript.

## Competing interests

The authors declare no competing interests.
