## [Peer Review File · Communications Biology]

Reviewers' comments:

Reviewer #1 (Remarks to the Author):

The research deals with an interesting subject of prokaryotic bioindicators for water quality assessment. The data set is large and might be very interesting, however, I was very disappointed by its use, analysis and writing. The paper is written at many places very generally without pointing to the important ideas, hypotheses, practical needs etc. See some examples below. Lots of information is missing. Please, provide a map of sampling sites, simplified table of samples, replicates, sequence numbers etc. Add methods describing the metadata assessment.

Specific comments:

Abstract:

Be more specific, write about goals, methods and their fulfilment. Give directly the names of taxa which are redundant or which are good bioindicators.

Methods: The sequence analysis is described in detail, however, some things are missing such as how exactly (which package) were metadata included to the analyses, e.i. combined with sequence information. In contrast, no need to describe in detail the individual steps of e.g. CUTADAPT, just write, which packages you used for what type of analyses. Same for dada2, no need to describe what it does, no need to write that it merged sequences at the end etc., just that you used it for sequence analyses.

Introduction:

Too general.

49-58 Shorten. start with In recent,...omit legislation, continue with In Europe,...

64-66 Very imprecise. Prokaryotes were first inhabitants on Earth, they developed all metabolisms, those are not adaptations. Yes, they are adaptable but in a different sense. They also have different electron acceptors not only donors. It is too general anyway.

74--111 This is not a historical review of prokaryotic bioindication, write about important issues, not chronology.

129-135 Way too general.

136- Write about the study goals, not the JDS goals.

Results.

It is very strange to show data on bacterial phyla and genera. These are too broad and too narrow indicators, resp. The indicators should be sought just in the taxonomic levels inbetween because they have the ecology and precision relevance.

The results should describe the figures and tables, which they do not. Those descriptions occur partly in the discussion. The figures are too simplified and do not show anything interesting.

Discussion. The description of ecological-biological relevance of selected phyla and genera does not make any sense at these two levels. Phyla are too broad and genera are often not identified correctly in such large and variable data set. There are definitely many unknown taxa present in the data set and those are not correctly placed to lower taxonomy levels. More effort should be paid to comparison between observed prokaryotic bioindicators and other used indicators as suggested in the introduction.

Reviewer #2 (Remarks to the Author):

Communications Biology Review: Fontaine et al. 2022
Bacterial bioindicators for biological status classification along a continental river

General Comments:

Fontaine et al. presents a robust modeling approach to determining bacterial indicators in river systems. More specifically, the authors take advantage of long-term data collection of the Danube River, and use environmental and bacterial taxonomic data (16S amplicon sequencing) to determine specific ASVs that could be used as bioindicators for water quality and eutrophication. Using two different approaches, Fontaine et al. identified a common set of ASVs that showed high predictive power with regards to modeling of biological status and chlorophyll-a concentrations. Many 16S studies often discuss the potential for using such data to help drive ecological modeling, but (as the authors note), there are very few studies that incorporate such data into models. The authors thus present an important methodological study utilizing the predictive power of bacterial communities and I would recommend this manuscript for publication, with some revisions. Although authors do give some detail on their modeling approach, more explanation (and clarification) can be given. In addition, basic sequence metrics should be shared (total sequences, avg. no. of sequences per sample). I also suggest minor edits to figures. Overall, I think it is a novel well-presented study that can be used to inform further ecological modeling and help inform long-term monitoring programs.

Specific Comments:

Lines 128-143: Adding in a sampling map and (brief) explanation of how samples were collected (e.g. middle, left, right transects), number of samples, months/years collected would be beneficial. Yes, this information can be found in the papers cited, but having at least some of this data in this study gives more context and can help give more meaning to results. For example, it is a bit confusing in the results when different transects are mentioned (middle vs. left, right etc), but sample collection was not explained in the methods.

Line 224: Adding a supplementary table of sequencing results would be helpful. What was the total number of sequences used? What was the average number of sequences per sample? Were samples normalized by sequencing depth?

Lines 238-241: Not sure what this sentence is trying to say, how were variables with missing data treated within the model? Maybe the word "which" doesn't belong here?

Methods, in general: Modeling methodology should be more clearly defined. I think breaking up into separate subsections to explain the methods for the "Spatio-temporal" modeling and the "Presence/variation" modeling separate (just like how the results are divided).

Figure 1: Can a supplementary table also be made of metadata? Separated by location (Up vs downstream)? This would make it easier to understand trends in the environmental data and how they relate to modeling results

Figures 2 and 3: Make the environmental variable vectors a different color. I can hardly see the light gray, and didn't even notice them at first!

Figure 4: Make the X and Y axis the same scale

Figure 6: Legend in combined PDF figure does not match that of Figure 6 uploaded separately. I think the latter legend is a better description

Reviewer #3 (Remarks to the Author):

The manuscript of Fontaine and coworkers deals with the identification of possible bacterial indicators of environmental quality in the Danube river, benefiting of a long term molecular data available from a international consortium.

The approach is very interesting and I fully agree with the necessity to explore prokaryotic diversity or functionality to get more efficient bioindicators of the biological status of freshwaters. Hence, I really welcome this manuscript. However, from my point of view there are some criticisms that should be addressed, or better explored in the discussion.

1- First of all the authors chose to check for taxonomy rather than functionality, with the background hypothesis that taxonomy is rather correlated to functionality. Of course the Danubian 16S rRNA database allows only this kind of search, instead of a full functional gene analysis. But this is a database defect, and would be addressed in the discussion. Especially considering that 16S rRNA is becoming a weaker taxonomy identifier against genome reconstruction. Moreover, several bacterial species may have quite different behaviour thanks to plasmids and other mobile elements. HGT could have an impact on the distribution of different taxa, but this has never been considered in the manuscript.

2- I am not sure regarding how water samples have been collected. On stable rivers, microorganisms on the top layer of the water are different from those few-cm down the layer, and different from the deep part of the river. I am pretty sure that authors took care of this, but it is hard to judge. Referring to the ICDPR is quite confusing since it is a complex homepage with too many popular information. Search for more specific technical information was challenging. I would ask authors to provide more basic information related to the sample collection.

3- In lines 483-490 there is an ecological hypothesis giving reason to the biogeographical distribution of phyla along a river. On a hand, this is quite interesting and, personally, I was not aware of a such macro-distribution. But the hypothesis behind it could be several, encompassing not only C availability from the riparian zone, but also the water speed, the availability of microaggregates that allow a stable biofilm, the water chemistry (difference in metal or ions, presence of tannins...), and many other. So this should be explored and, if possible, deepened.

4- Authors removed DNA sequences from chloroplasts. This is normal. But did this affect the analysis of cyanobacteria? Please, discuss this point.

5- In the abstract, lines 35-37 were too general. Please, provide more information.

6- Lines 66-68 show a quite common knowledge. However, most of bacterial cells are quiescent and not responsive against an environmental change, until the environmental conditions allow their metabolic activities, or affect their integrity. How does this affect the authors' conclusions?

7- In lines 327, 330 and elsewhere, authors referred to abundance. How were they secure of quantification through 16S rRNA Illumina sequencing, considering the technical biases? Moreover, 16S rRNA quantification should be normalized with the number of rRNA operons within individual cells. Maybe this is not a big issue when speaking at a higher taxon level, but at genus level the operon copy number may affect the final quantification. Did author consider this point too?

Reviewers' comments:

Reviewer #1 (Remarks to the Author):

The research deals with an interesting subject of prokaryotic bioindicators for water quality assessment. The data set is large and might be very interesting, however, I was very disappointed by its use, analysis and writing. The paper is written at many places very generally without pointing to the important ideas, hypotheses, practical needs etc. See some examples below. Lots of information is missing. Please, provide a map of sampling sites, simplified table of samples, replicates, sequence numbers etc. Add methods describing the metadata assessment.

A map of the sampling sites and a table of samples have been added in supplementary materials. Metadata assessment was carried out under the auspices of the ICPDR; we used the metadata as provided without further examination.

Specific comments:

Abstract:

Be more specific, write about goals, methods and their fulfilment.

We highlight the problem L29-31 "Despite the importance of bacteria in aquatic ecosystems and their predictable diversity patterns across space and time, biomonitoring tools for status assessment relying on these organisms are widely lacking."

And mention the methods L32 "Here, we used metabarcoding in combination with multivariate statistics and machine learning to identify bacterial bioindicators ..."

We also provide a refined final sentence to summarize the outcome of our study. L44-46 "Our study shows bacterial indicators are robust for biological status classification and therefore should be considered for integration into bioassessment schemes in global conservation and restoration efforts."

Give directly the names of taxa which are redundant or which are good bioindicators.

We have added that ASVs from the genera *Fluviicola*, *Acinetobacter*, *Flavobacterium* and *Rhodoluna* are good indicators. See L40.

Methods: The sequence analysis is described in detail, however, some things are missing such as how exactly (which package) were metadata included to the analyses, e.i. combined with sequence information. In contrast, no need to describe in detail the individual steps of e.g. CUTADAPT, just write, which packages you used for what type of analyses. Same for dada2, no need to describe what it does, no need to write that it merged sequences at the end etc., just that you used it for sequence analyses.

We have changed the methods section by providing additional detail on the statistical analyses and removed some details on sequence processing. Still we kept essential information that is required for reproducing the sequence data processing.

Introduction:

Too general.

We have substantially modified the introduction to make it less general. For example, we have shortened details on bacteria and their role in the environment.

49-58 Shorten. start with In recent,...omit legislation, continue with In Europe,...

The section has been shortened in the requested manner.

64-66 Very imprecise. Prokaryotes were first inhabitants on Earth, they developed all metabolisms, those are not adaptations. Yes, they are adaptable but in a different sense. They also have different electron acceptors not only donors. It is too general anyway.

We pruned the introduction from general statements about prokaryotic organisms and their metabolic adaptations to environmental pressures.

74--111 This is not a historical review of prokaryotic bioindication, write about important issues, not chronology.

We removed superfluous information about previous work on water quality-related bioindicators, only retaining key elements of previous research to contrast our findings against.

129-135 Way too general.

We removed superfluous information about the JDS project in general and added information about the collection of materials which went into generating data used for this manuscript.

136- Write about the study goals, not the JDS goals.

We removed JDS goals from the manuscript and updated study goals in the introduction. These are now stated as such (see L99-109) "Here we seek to establish a method for identifying reliable bacterial bioindicators for characterization and prediction of ecological patterns such as ecological status and water quality of the Danube River. We hypothesize the mobility of organisms within the river makes upstream community composition informative for downstream ecological outcomes such as ecological status from Saprobic index and chlorophyll *a* concentration. In addition, we applied multivariate statistics to detect suitable bioindicators based on the prevalence and variance of prokaryotic taxa (3) associated with environmental drivers, as a snap-shot approach not accounting for the spatial dependency of the sites. Finally, we compared the results from the snap-shot and spatio-temporal approaches to evaluate the efficiency and transferability of bacterial bioindicators for the assessment of river ecosystem's biological status as part of conservation and restoration efforts."

Results.

It is very strange to show data on bacterial phyla and genera. These are too broad and too narrow indicators, resp. The indicators should be sought just in the taxonomic levels inbetween because they have the ecology and precision relevance.

The data shows the lowest resolution of the taxonomic annotation. We added taxonomic levels on the x axis for figure 5 to make visualization easier and more informative. Nodes on the dendrogram represent taxonomic levels. We show data on the various taxonomic levels to highlight that both

broad and narrow indicators provide information. This has been highlighted before by for example Rimet and Bouchez (2012) for diatoms and Jones (2008) for benthic macroinvertebrates. Here we show this for bacteria.

The results should describe the figures and tables, which they do not. Those descriptions occur partly in the discussion. The figures are too simplified and do not show anything interesting.

We have moved figure 1 to supplementary material. Figure 5 was enhanced with indication of node position as taxonomic level as well as histograms of ASV counts to put into perspective their relevance in terms of information content. Adding ASVs in the network figures (Fig. 4) makes them quickly unreadable from the sheer number of graph edges. We have also adjusted results and discussion accordingly.

Discussion. The description of ecological-biological relevance of selected phyla and genera does not make any sense at these two levels. Phyla are too broad and genera are often not identified correctly in such large and variable data set. There are definitely many unknown taxa present in the data set and those are not correctly placed to lower taxonomy levels. More effort should be paid to comparison between observed prokaryotic bioindicators and other used indicators as suggested in the introduction.

For the variance/prevalence analyses, we have added results for order and family to provide taxonomically relevant ecological patterns. This does not apply to the spatio-temporal approach as the analyses are run on specific ASVs, whereas the color coding for figure 5 matching bacterial classes is used for ease of visualization.

Cited papers on previous bioindicator research (Fortunato et al. 2013) used phylum, family and genus level, which motivated our choice of these taxonomic levels for ease of comparison between the present study and previous ones. Some authors used mixed taxonomic levels in clades presented in a single analysis. This is explainable by assumptions of ecological relevance for specific clades. It runs against our spatio-temporal approach, which is to proceed without assumptions and see which ASVs give the best signal. Grouping these ASVs by taxonomic level becomes of secondary importance in this context. Furthermore, ecological function does not necessarily follow taxonomy and as we lack genomic data to solve these issues in the present study, we believe the limitations imposed by 16S data on interpreting ecological meaning from taxonomic assignments mandate great caution on our part while diminishing the relevance of accurate identification and grouping.

Reviewer #2 (Remarks to the Author):

Communications Biology Review: Fontaine et al. 2022

Bacterial bioindicators for biological status classification along a continental river

General Comments:

Fontaine et al. presents a robust modeling approach to determining bacterial indicators in river systems. More specifically, the authors take advantage of long-term data collection of the Danube River, and use environmental and bacterial taxonomic data (16S amplicon sequencing) to determine specific ASVs that could be used as bioindicators for water quality and eutrophication. Using two

different approaches, Fontaine et al. identified a common set of ASVs that showed high predictive power with regards to modeling of biological status and chlorophyll-a concentrations. Many 16S studies often discuss the potential for using such data to help drive ecological modeling, but (as the authors note), there are very few studies that incorporate such data into models. The authors thus present an important methodological study utilizing the predictive power of bacterial communities and I would recommend this manuscript for publication, with some revisions. Although authors do give some detail on their modeling approach, more explanation (and clarification) can be given.

We have expanded the description on the modeling in the methods section. L216-314

“Statistical analyses and plot generation were conducted in R version 4.0.2 (2019-12-12; R Core Team, 2014). From the ASV table, we created three datasets, clustering the ASVs on phylum, class and genus levels. For each of the three datasets we calculated the prevalence (occurrence frequency at each sampling site) and coefficient of variation (standard deviation of taxon abundance divided by the mean) (3). These two parameters were plotted against each other in a scatter plot to detect taxa with the highest variance occurring more often across all sampling sites, as these might represent suitable biological indicators.

To identify the environmental drivers behind bacterial community assembly, we selected specific variables among the metadata ('pH', 'Electric Conductivity', 'water temperature', 'River km/Distance to mouth', 'Ntot', 'Chl a', 'DOC', 'Total Coliforms (LOG10(x + 1)', 'BacHum (LOG10(x + 1)', 'AllBac (LOG10(x + 1)', 'E.coli (LOG10(x + 1)') (Supplementary fig. S2). Missing observations were imputed with the *mice* function from the 'namesake' R package (version 3.13.0) and the VIM package (version 6.2.2), using the Predictive Mean Matching (PMM) imputation approach (25). Relationships between beta diversity and chlorophyll *a* were investigated in order to characterize the extent of linearity along this gradient and look for possible thresholds associated with shifts in microbial community composition. The analysis was performed using pairwise bacterial community composition (BCC) Bray-Curtis distances along the chlorophyll *a* gradient according to Fontaine *et al.* (26).

Prevalence/variance analysis of the microbial community structure

By using a step forward selection model (*vegan* version 2.6-2), we selected the variables most related to the Hellinger-transformed Bray-Curtis distance matrix for each of the three bacterial taxonomic levels. The aim of forward selection was to identify environmental parameters covarying with community composition. Next, the most informative variables were tested for multicollinearity by using the *Variance Inflation Factor* (VIF) value and tolerance value. Those variables showing a VIF value above 5 were excluded from further analyses. A distance-based Redundancy Analysis (dbRDA) was performed by using the Hellinger-transformed Bray Curtis matrix for each of the three taxonomic levels by only including the most prevalent and variable taxa with the above selected metadata to identify specific relationships amongst phyla, classes or genera with particular environmental drivers.

Spatio-temporal identification of bacterial bioindicators

Biological status-related variables (chlorophyll *a* concentration as a proxy for eutrophication and classification of saprobity according to the SI) for downstream sites were predicted using information from upstream bacterial community composition. The identification of ASVs informative for biological status prediction was performed in two main steps. At first, we created an unsupervised random forest (RF) model (*randomForest* version 4.6-14) in order to gain knowledge on

the latent structure of the sites based on ASVs, in order to prefilter the ASV table due to the following screening step being computationally demanding. Without knowledge of the true structure within the data, a grid search optimization of parameters for the unsupervised RF model was performed on a supervised model where the response variable was chlorophyll *a* concentration, and the explanatory data was the Hellinger-transformed ASV abundance table. The lowest mean squared error (MSE) of these supervised learning models was obtained with a combination of $mtry=1600$ and $ntree=30$ with a 0.8/0.2 dataset split for training and testing sets. Using these parameters, the unsupervised RF model was trained on the same Hellinger-transformed ASV abundance table. From the unsupervised model output, candidate ASVs for downstream analysis were selected based on exceeding a threshold of 0 in percentage increase of mean squared error (%IncMSE).

The second main step in the ASV screening process was to run, for each ASV in decreasing order of %IncMSE, a grid search for the optimal combination of steps (i.e. number of upstream sites) to use for (i) transformation and (ii) lag using XGboost (sci-kit learn implementation of XGboost version 1.3.3). In other words, the objective of the grid search is to find the two optimal shifting frames of number of sites for each ASV used for predicting response variables. The transformation in this case is scaling and centering of ASV abundances, while lag is the number of observations needed from upstream sites for the prediction at a given downstream site. The criterion for selecting the best combinations of values for transformation and lag was the model R squared on the test set for chlorophyll *a* and accuracy percentage for biological status classification. The design matrix for the ASV screening initially contains sampling ID, transect code (left, right and middle) and distance to river mouth. These metadata variables are included to account for the continuity of each longitudinal transect in terms of laminar flow dynamics as well as travel time in the case of distance to river mouth. Upon yielding an improved model R squared, an ASV is added to the design matrix in its optimal transformation and lag combination. The ASV screening process was run 1000 times, for both water quality variables, with random ASV order and random combinations of XGboost hyperparameters. However, for the classification based on the SI, two observations presented high biological status and first appeared at 550 km from the river mouth (Supplementary fig. S3), which is past the first 80 % of observations, by order of distance from the Danube source. It is impossible to evaluate model accuracy if the test set contains variable classes which are not present in the training set. Thus, in order to include observations of all variable classes in the training set, the training and testing split of the data for XGboost models (both for chlorophyll *a* concentration and biological status classification) was set at 0.85 and 0.15 respectively, with the split occurring after the first 85% of observations.

The model outputs for the ASV screening for biological classification were filtered to keep only those presenting an accuracy of 100 %. In the case of chlorophyll *a*, the model yielding the highest R squared was kept. The model outputs were turned into a presence/absence table with individual models as rows and ASVs as columns for water quality classification. The resulting table was then used for a network analysis, performed using R library igraph (version 1.2.4.2). Co-occurrence and co-exclusion networks were used to investigate information content and redundancy between ASVs. Functional redundancy between individual taxa, in the sense of information content translating into model predictive power, was here interpreted as ASVs which are never present together in any individual screening process output. The logic behind this assumption is that ASVs are mutually exclusive if they contain shared predictive information. As the screening process will pick up the first ASV of a set containing a given share of information, the following ones from that same set would be left out since they would not improve the model.

In opposition to co-occurrence where there is certainty about said relationship, the same cannot be assumed from co-exclusion found here since it could result from the limited number of random permutations of ASVs screened together. Nevertheless, a small number of ASVs appeared in most of the screening process outputs, suggesting a semblance of saturation in their sampling and thus that their co-exclusion relationships are accurate. The selection of ASVs kept for this co-exclusion analysis was thus set at the 95th percentile by number of occurrences. A dendrogram was then built with all the ASVs yielding the best models for eutrophication and water quality to visualize taxonomic overlap between predictors for both variables. The hierarchical clustering was performed on a distance matrix computed from the taxonomy table.”

In addition, basic sequence metrics should be shared (total sequences, avg. no. of sequences per sample). I also suggest minor edits to figures.

We have added basic sequence information to the supplementary material, supplementary table 1. We also modified the figures of the manuscript, see details below.

Overall, I think it is a novel well-presented study that can be used to inform further ecological modeling and help inform long-term monitoring programs.

Specific Comments:

Lines 128-143: Adding in a sampling map and (brief) explanation of how samples were collected (e.g. middle, left, right transects), number of samples, months/years collected would be beneficial. Yes, this information can be found in the papers cited, but having at least some of this data in this study gives more context and can help give more meaning to results. For example, it is a bit confusing in the results when different transects are mentioned (middle vs. left, right etc), but sample collection was not explained in the methods.

We removed superfluous information about the JDS project in general and added information about the collection of materials which went into generating data used for this manuscript.

We have added a map of the sampling sites (Supplementary figure 1).

Line 224: Adding a supplementary table of sequencing results would be helpful. What was the total number of sequences used? What was the average number of sequences per sample? Were samples normalized by sequencing depth?

We have added a supplementary sample table (Supplementary table 1) with sequence information. Samples were not normalized by sequencing depth.

Lines 238-241: Not sure what this sentence is trying to say, how were variables with missing data treated within the model? Maybe the word “which” doesn’t belong here?

We have modified the introduction and this sentence is now highly modified.

Methods, in general: Modeling methodology should be more clearly defined. I think breaking up into separate subsections to explain the methods for the “Spatio-temporal” modeling and the “Presence/variation” modeling separate (just like how the results are divided).

We agree with the reviewer and have added section titles in the methods section.

Figure 1: Can a supplementary table also be made of metadata? Separated by location (Up vs downstream)? This would make it easier to understand trends in the environmental data and how they relate to modeling results

We have moved figure 1 to supplementary material and expanded information on the sites with a map and table for selected metadata variables. Supplementary figure S3 provides an overview of the trends for ecological status and chlorophyll a by location along the river.

Figures 2 and 3: Make the environmental variable vectors a different color. I can hardly see the light gray, and didn't even notice them at first!

We agree with the reviewer and changed the vector's color to black. Contrast should now be clearer.

Figure 4: Make the X and Y axis the same scale

The axes in figure 4 were made equal.

Figure 6: Legend in combined PDF figure does not match that of Figure 6 uploaded separately. I think the latter legend is a better description

The discrepancy has been fixed. Now this is figure 5 and it has been modified to address comments. The legend from the separately uploaded figure is now used for both versions.

Reviewer #3 (Remarks to the Author):

The manuscript of Fontaine and coworkers deals with the identification of possible bacterial indicators of environmental quality in the Danube river, benefiting of a long term molecular data available from an international consortium.

The approach is very interesting and I fully agree with the necessity to explore prokaryotic diversity or functionality to get more efficient bioindicators of the biological status of freshwaters. Hence, I really welcome this manuscript. However, from my point of view there are some criticisms that should be addressed, or better explored in the discussion.

1- First of all the authors chose to check for taxonomy rather than functionality, with the background hypothesis that taxonomy is rather correlated to functionality. Of course the Danubian 16S rRNA database allows only this kind of search, instead of a full functional gene analysis. But this is a database defect, and would be addressed in the discussion. Especially considering that 16S rRNA is becoming a weaker taxonomy identifier against genome reconstruction. Moreover, several bacterial species may have quite different behaviour thanks to plasmids and other mobile elements. HGT could have an impact on the distribution of different taxa, but this has never been considered in the manuscript.

We consider that metabarcoding is a cheaper and more high throughput approach, it is currently more suitable for bioindicator analyses. Furthermore, the high variability among genomes does not allow for broad application of identified bioindicators. Broader groups such as genera as defined by amplicon sequencing are more likely universally distributed and thus suitable for bioindicator analyses. This is now discussed in more detail in the manuscript.

An additional outstanding issue with genomic information is annotation since gene ontology is not settled. Still, one can expect less COG genes in a dataset than the number of ASVs, which makes it easier to screen them for ecological information relevance. The feasibility of environmental genomic studies depends heavily on funding to generate useable datasets. Currently, existing funding schemes do not allow to build a comprehensive global genomic dataset for a system such as the Danube.

Sequencing data type and suitability for functional analyses is mentioned in the discussion (lines 459-475). We agree with the reviewer that full functional gene information is by far preferable to 16S rRNA as function ultimately translates into ecological outcomes and taxonomy is at best a noisy proxy. In the meantime however, there is useable information that can be derived from 16S community composition as has been presented in this study and this is why it amounts to excessive caution to wait for a complete functional dataset for the Danube river to begin work on how to identify bioindicators. It is reasonable to assume the analyses performed here are fully transferable to functional information datasets.

We did not have information on mobile elements, plasmids, conjugation and other mechanisms involved in HGT to infer on their contribution to community assembly in this study. We agree their role ought to be meaningful but could only observe signals from the data we had at hand. Clear signals came through despite the obvious weaknesses of 16S rRNA data. Niche overlap and the resulting competition within a same guild dilutes the functional information over its various member taxa while ultimately never revealing what that functional information is. In other words, taxonomic data is a noisy form of functional information and is in theory dimensionally reducible to a table of metabolic pathways.

2- I am not sure regarding how water samples have been collected. On stable rivers, microorganisms on the top layer of the water are different from those few-cm down the layer, and different from the deep part of the river. I am pretty sure that authors took care of this, but it is hard to judge. Referring to the ICDPR is quite confusing since it is a complex homepage with too many popular information. Search for more specific technical information was challenging. I would ask authors to provide more basic information related to the sample collection.

More detail on the sample collection is now provided in the methods. L126-131

“Water samples for the microbial community were collected by hand from the epilimnion in 1L flasks, at the same time the physico-chemical parameters of the water were also measured with hand probes. Macroinvertebrates samplings were performed using a Multi Habitat Sampling approach, where different parts of the riverbed were disturbed, and the macroinvertebrates collected with a net. Left and right sides of the river, together with its center, were sampled at 60 locations.”

3- In lines 483-490 there is an ecological hypothesis giving reason to the biogeographical distribution of phyla along a river. On a hand, this is quite interesting and, personally, I was not aware of a such macro-distribution. But the hypothesis behind it could be several, encompassing not only C availability from the riparian zone, but also the water speed, the availability of microaggregates that allow a stable biofilm, the water chemistry (difference in metal or ions, presence of tannins...), and many other. So this should be explored and, if possible, deepened.

These are planktonic samples – only the diffuse part of the planktonic community was considered – microaggregates allowing stable biofilms were not taken into account. We agree with reviewer 3 that

additional variables can be relevant to explain biogeographical distribution of phyla along a river, but the variables mentioned here were unavailable for this study. The macro-distribution and the ecological mechanisms driving the complex patterns have been explored in multiple previous publications see for example Savio et al. 2015, Read et al. 2015, etc.

4- Authors removed DNA sequences from chloroplasts. This is normal. But did this affect the analysis of cyanobacteria? Please, discuss this point.

We removed chloroplast sequence from the final ASV table by removing all chloroplast sequences at order level as is implemented in the SILVA database. To review the chloroplast and cyanobacterial sequences from the SILVA database goes beyond the resources of our project.

5- In the abstract, lines 35-37 were too general. Please, provide more information.

We have added “Bacterial beta-diversity dynamics followed environmental gradients and the observed associations highlighted potential bioindicators for ecological outcomes”.

6- Lines 66-68 show a quite common knowledge. However, most of bacterial cells are quiescent and not responsive against an environmental change, until the environmental conditions allow their metabolic activities, or affect their integrity. How does this affect the authors' conclusions?

We could not examine activity levels for the various taxa as the required materials were not collected during the sampling campaign; we only looked at community composition and which taxa provided the best signals regarding ecological outcomes, regardless of the metabolic state of individual ASVs found to be relevant.

7- In lines 327, 330 and elsewhere, authors referred to abundance. How were they secure of quantification through 16S rRNA Illumina sequencing, considering the technical biases? Moreover, 16S rRNA quantification should be normalized with the number of rRNA operons within individual cells. Maybe this is not a big issue when speaking at a higher taxon level, but at genus level the operon copy number may affect the final quantification. Did author consider this point too?

The reviewer points out a common well known issue with sequencing data. We considered this point and now use term such as “relative abundance” and “contribution” to align with the compositional character of sequencing data. Copy number variation is another major issue and affects compositional data. In particular variations within the various taxonomic groups can obscure outcomes of statistical analyses – we would argue that variations might be larger at the phylum than genus level. However, it can also be argued that besides this issue (or noise) we were able to identify bioindicators.

Limitations are now mentioned in the discussion.

“This study highlights the potential of the identification of bacterial bioindicators for assessing the biological status of river ecosystems from bacterial metabarcoding data, despite methodological constraints such as copy number variations (27), PCR biases (28), compositional character of the sequencing data (29) as well as other limitations (30).”

Reviewers' comments:

Reviewer #1 (Remarks to the Author):

The MS as it was written before might be very interesting because of a large dataset, which was used and quite precise analyses. However, it is still very far from acceptable due to places with misunderstanding such as in the introduction about predictability of bacterial diversity. What is meant by that? As far as I know it is highly unpredictable. The abstract is still very general, does not specifically address the results and the whole MS is like that. E.g. where the relationship to macroinvertebrates indication is presented? What is a biological statues? What is the biological status of the individual sites? You named several indicator genera but what exactly they indicate?

Or by constant mixing of microbial, prokaryotic and bacterial in commenting results and in the discussion. The MS is clearly about bacteria and it is not possible to go back and forth with using these three terms, inadequately. Many more things along this line could be named. Although the methods were describe in a better way they still do not provide sufficient information for someone to repeat the work. The results are completely confusing, there are too many figures, some of a very poor quality i.e. too simplified such as those presenting RDA with linear vectors (where are the individual samples on the figures) or too complicated without any clear explanation such as Fig. 3 and Fig. 5. Please bear in mind that no one can spend a long time with learning what the figures mean, they by definition need to be self sustaining. They should demonstrate not the extend of your work but clear picture of conclusive results.

There are still also some technical problems I am not sure about. Firstly, you should use ASVs as variables in your calculations and in your presentation of figures, not taxonomic groups to which they were assigned. That you need to do but only after your analyses. It is really very questionable if individual bacterial taxa separated from their communities mean something because there is an endless number of interactions, which may condition the occurrence of some taxa much more than local conditions. If you want to keep the analyses like this, you still need to discuss this work, preferably add some analyses about co-occurrence patterns. Redundancy in bacterial taxa is a very questionable concept, remember e.g. paradox of plankton, the same applies here, why there would be so many taxa if they did not have their niche and therefore a specific function in the community. Something along this line should be discussed with your spatio-temporal results etc. I would recommend most of all to ask some experineced microbial ecologist to help you with these issues and put the results to a much firmer ground that as currently presented.

Reviewer #2 (Remarks to the Author):

Communications Biology Review: Fontaine et al. 2023, Revised
Bacterial bioindicators for biological status classification along a continental river

General Comments:

In the revised version on the manuscript, Fontaine et al. again presents a two modeling approaches to determining bacterial indicators in river systems. Specifically, the use long-term data collection of the Danube River and use environmental and 16S amplicon sequencing data to determine specific ASVs that could be used as bioindicators for water quality and eutrophication. In the previous version of the manuscript, I noted that more information was needed on sample collection as well as further explanation on the two different modeling approaches used. In their rebuttal letter, the authors addressed these concerns and edited the text accordingly. However, further clarification and editing of methodology is still needed (see specific comments). I also think the authors get caught up in explaining the caveats to the methods but do not capitalize on the novelty of using bacterial ASVs as bioindicators, I'd like to see some text in the discussion about application of these approach and how

it actually could be used in ecological monitoring and restoration efforts. Overall, this is a stronger paper for which I would recommend for publication, with revisions.

Specific Comments:

Abstract: Include more results, less explanation. Results for presence/variance? Need stronger last sentence of abstract.

Lines 32-35: Simplify sentence, this is confusing. Perhaps remove last lines, end at "for existing biological status".

Lines 41-44: Clarify sentence, don't need part about redundancy, too detailed. Just that modeling showed accurate prediction of biological status with 2-3 ASVs.

Introduction:

Line 86: Write out morphological taxa identification (not morpho-taxonomy)

Line 109: Typo. Should be restoration

Methods:

Line 130 and throughout: Be consistent with naming transects. Here they are listed as right, left, center. Later in the manuscript "center" is called "middle". Please fix.

Line 144: SI index is mentioned on Line 103, should define there too.

Lines 138-150: Edit. Don't need long explanation, just say SI and chlorophyll were used for validation of modeling.

Line 171-172: Redundant, already stated on Line 156.

Line 204-205: Simplify. Chimeras were removed using the 'removeBimeraDenovo' function in dada2.

Lines 249-314: I do appreciate the expanded explanation of the spatial-temporal modeling, but I think authors provide too much unnecessary detail here. Some could be moved to a methods supplement. It is very hard, especially towards the end of this section, to follow exactly what the authors are saying. Is there a simpler way to summarize this for the general (non-modeling expert) reader.

Results:

Lines 337-342, and Figure 3: First I'm not sure the point of the paragraph, other than trying to explain (not very well), why the R2 value between BCC and Chl A was low? I also think Figure 3 is confusing....it necessary to be in manuscript? A rule of a good figure is that it should be easy to interpret the main takeaway, this is not true for Fig. 3. Add labels directly on Figure? Move to supplement?

Discussion:

485-487: Expand here. Cite papers that have used similar modeling, explain how this study builds/is different. It is important to capitalize on novelty of study.

Lines 505-525: Very repetitive paragraph, revise.

Reviewer #3 (Remarks to the Author):

The manuscript's authors solved the issues I raised in the first revision and I consider it suitable for the publication.

Reviewers' comments:

The MS as it was written before might be very interesting because of a large dataset, which was used and quite precise analyses. However, it is still very far from acceptable due to places with misunderstanding such as in the introduction about predictability of bacterial diversity. What is meant by that? As far as I know it is highly unpredictable.

There is a lot of literature on the predictability of microbial communities including bacterial diversity; i.e. <https://www.pnas.org/doi/10.1073/pnas.0602399103>
<https://ami-journals.onlinelibrary.wiley.com/doi/full/10.1111/1462-2920.12886>
<https://www.pnas.org/doi/abs/10.1073/pnas.1917265117>
https://www.science.org/doi/full/10.1126/science.aat1168?casa_token=ucls-0gUBxQAAAAA%3AHL40HrY4I-6fvMS4sbwVa3nGoAiYOAf7aUQSAgHxyUlwzrVXjoxHcHNYb_TZbbmbJSNjYb0YZJP3qM
<https://www.nature.com/articles/ismej200764>
etc.

Here we provide specific references to microbial communities in river systems and introduce these publications and their findings in the introduction.

Fortunato CS, Eiler A, Herfort L, Needoba JA, Peterson TD, Crump BC. Determining indicator taxa across spatial and seasonal gradients in the Columbia River coastal margin. *ISME J.* 2013 Oct;7(10):1899–911.

Aylagas E, Borja Á, Tangherlini M, Dell'Anno A, Corinaldesi C, Michell CT, et al. A bacterial community-based index to assess the ecological status of estuarine and coastal environments. *Mar Pollut Bull.* 2017 Jan 30;114(2):679–88.

Savio D, Sinclair L, Ijaz UZ, Parajka J, Reischer GH, Stadler P, et al. Bacterial diversity along a 2600 km river continuum. *Environ Microbiol.* 2015;17(12):4994–5007.

The abstract is still very general, does not specifically address the results and the whole MS is like that. E.g. where the relationship to macroinvertebrates indication is presented? What is a biological status? What is the biological status of the individual sites? You named several indicator genera but what exactly they indicate? This manuscript does not intend to describe individual sites biological status and we do not think that this is of interest to the readers. As emphasized already in the abstract (also throughout the manuscript) the indicator taxa indicate biological status as assessed by macroinvertebrates as well as predict chlorophyll *a* concentration.

Or by constant mixing of microbial, prokaryotic and bacterial in commenting results and in the discussion. The MS is clearly about bacteria and it is not possible to go back and forth with using these three terms, inadequately. Bacteria are prokaryotes and microbes, thus these terms can be used interchangeably to a certain extent. We have gone through the manuscript and changed this a few times so that it could not be interpreted as inappropriate.

Many more things along this line could be named. Although the methods were describe in a better way they still do not provide sufficient information for someone to repeat the work.

This is hard for us to implement as no detail is provided on which parts should be expanded. Methods that have been used including the molecular methods were appropriately referenced when possible thus no further detail is provided in these cases. Bioinformatic processing follows the dada2 analysis procedure as outlined in the tutorial and settings that are different from default settings are reported. Methods to assess environmental properties are given in the ICPDR report and refer to standardized methods used in water quality assessment. Data files used in modeling are now provided on github so is all code used in this manuscript. https://github.com/alper1976/danube_indicators

The results are completely confusing, there are too many figures, some of a very poor quality i.e. too simplified such as those presenting RDA with linear vectors (where are the individual samples on the figures) or too complicated without any clear explanation such as Fig. 3 and Fig. 5. Please bear in mind that no one can spend a long time with learning what the figures mean, they by definition need to be self sustaining. They should demonstrate not the extend of your work but clear picture of conclusive results.

One of the most powerful aspects of RDA is the **simultaneous visualization** of response and explanatory variables (i.e. species and environmental variables), as given in our figures. To make the figure less complicated we did not include the sites (samples) in the plot. To guide the reader as suggested by the reviewer we have added additional information to the captions of figures 1-3 and 5. Example for figures 3 and 5:

“**Figure 3.** Threshold analysis for microbial beta diversity (Bray–Curtis distances) along the chlorophyll *a* gradient. The color scale represents Bray-Curtis distance values computed between sites. The axes stand for chlorophyll *a* concentration (mg/L) at each site. In panel a) the observed beta diversity patterns are presented along the chlorophyll *a* gradient, as modeled using XGboost. In panel b) a hypothetical relationship is represented where dissimilarity between communities increases linearly as a function of the difference in chlorophyll *a* between sites. The mean value of the response surface in panel a) can be treated as the baseline beta-diversity across all sites. Data points with values below the mean represent higher similarity between sites; likewise, higher values

represent lower similarity. To interpret the response surfaces of observed values, one may begin by looking at a point bordering the diagonal and then follow a line of points further up on the chlorophyll a axis. Here, a “ridge” indicates a chlorophyll a concentration to be a likely threshold from which the shift in bacterial community composition is greater than average. In the same manner, a “valley” indicates a chlorophyll a concentration likely located on an interval of the chlorophyll-a gradient along which bacterial communities do not shift substantially. More details on the XGboost approach and its interpretation are given in Fontaine et al. (27).”

“**Figure 5.** Phylogenetic analysis of the bioindicator taxa identified for the best predictive models for biological status classification and eutrophication (chlorophyll a concentration). Colors represent bacterial classes while symbols represent the variable for which an ASV was identified as a predictor. The horizontal position of nodes in the dendrogram indicates at which taxonomic level they occur, marked with dotted lines from Kingdom to genus. The histogram represents the relative frequency of each ASV’s occurrence in model outputs.”

There are still also some technical problems I am not sure about. Firstly, you should use ASVs as variables in your calculations and in your presentation of figures, not taxonomic groups to which they were assigned. That you need to do but only after your analyses.

ASVs were used in our calculations as well but here we only represent taxonomic groups. ASVs will not be informative to the reader while taxonomic assignments are, thus each ASV is represented with its taxonomic assignment and not an arbitrary number.

It is really very questionable if individual bacterial taxa separated from their communities mean something because there is an endless number of interactions, which may condition the occurrence of some taxa much more than local conditions. If you want to keep the analyses like this, you still need to discuss this work, preferably add some analyses about co-occurrence patterns.

It is essential to what we show that individual taxa contain information to predict status and as such environmental conditions. The comment by the reviewer questions our entire manuscript and the outcome of our modeling approach. Our analysis strongly argues that our results show that individual taxa can be used irrespective of the interactions. This is now mentioned in the discussion.

Redundancy in bacterial taxa is a very questionable concept, remember e.g. paradox of plankton, the same applies here, why there would be so many taxa if they did not have their niche and therefore a specific function in the community. Something along this line should be discussed with your spatio-temporal results etc. I would recommend most of all to ask some experienced microbial ecologist to help you with these issues and put the results to a much firmer ground than as currently presented.

We are not sure about how to address the rather rude parts of this comment implying that we are not experienced microbial ecologists ☺.

We are well aware that the paradox of the plankton and redundancy theory are not competitive concepts – they address the same issue and provide complementary explanations for the high diversity in seemingly uniform aquatic environments – which is actually not the case when considering, for example, electron donor and acceptor concentration gradients in even a milliliter of water (meaning a drop of water is not uniform). In our manuscript redundant information content is used which could or could not correlate with actual ecological redundancy – as we do not have the data, the scope is not to resolve ecological redundancy. Hence discussing the massive literature on ecological redundancy is not the focus of this ms. We have now clarified this by changing the title of this section to “**Informational redundancy in bacterial indicators**”

Reviewer #2 (Remarks to the Author):

Communications Biology Review: Fontaine et al. 2023, Revised
Bacterial bioindicators for biological status classification along a continental river

General Comments:

In the revised version on the manuscript, Fontaine et al. again presents a two modeling approaches to determining bacterial indicators in river systems. Specifically, the use long-term data collection of the Danube River and use environmental and 16S amplicon sequencing data to determine specific ASVs that could be used as bioindicators for water quality and eutrophication. In the previous version of the manuscript, I noted that more information was needed on sample collection as well as further explanation on the two different modeling approaches used. In their rebuttal letter, the authors addressed these concerns and edited the text accordingly. However, further clarification and editing of methodology is still needed (see specific comments).

We have modified the methods section. See below

I also think the authors get caught up in explaining the caveats to the methods but do not capitalize on the novelty of using bacterial ASVs as bioindicators, I’d like to see some text in the discussion about application of these approach and how it actually could be used in ecological monitoring and restoration efforts.

We have added additional discussion about the potential applications and potential of the approach in the beginning of the discussion “**The disruptive potential of bacterial indicators for biological status classification**”

Overall, this is a stronger paper for which I would recommend for publication, with revisions.

Specific Comments:

Abstract: Include more results, less explanation. Results for presence/variance? Need stronger last sentence of abstract.

We have shortened some sentences as suggested by the reviewer and added a modified final sentence to the abstract.

"As such our models show that using a few bacterial ASVs from globally distributed genera allows for biological status assessment along river systems."

Lines 32-35: Simplify sentence, this is confusing. Perhaps remove last lines, end at "for existing biological status".

The sentence was changed to "Here, we used metabarcoding in combination with multivariate statistics and machine learning to identify bacterial bioindicators for existing biological status classification systems." As suggested by the reviewer.

Lines 41-44: Clarify sentence, don't need part about redundancy, too detailed. Just that modeling showed accurate prediction of biological status with 2-3 ASVs.

Sentence was modified as suggested to "The redundancy among bacterial bioindicators revealed mutually exclusive taxa, which allow accurate biological status modeling using as few as 2-3 ASVs."

Introduction:

Line 86: Write out morphological taxa identification (not morpho-taxonomy)

Fixed.

Line 109: Typo. Should be restoration

Fixed.

Methods:

Line 130 and throughout: Be consistent with naming transects. Here they are listed as right, left, center. Later in the manuscript "center" is called "middle". Please fix.

Fixed throughout the manuscript.

Line 144: SI index is mentioned on Line 103, should define there too.

The SI is just one index for macroinvertebrates thus we do not introduce it in the introduction.

Lines 138-150: Edit. Don't need long explanation, just say SI and chlorophyll were used for validation of modeling.

We have shortened this section by removing one and a half sentences.

Line 171-172: Redundant, already stated on Line 156.

This redundant sentence was removed.

Line 204-205: Simplify. Chimeras were removed using the 'removeBimeraDenovo' function in dada2.

Fixed.

Lines 249-314: I do appreciate the expanded explanation of the spatial-temporal modeling, but I think authors provide too much unnecessary detail here. Some could be moved to a methods supplement. It is very hard, especially towards the end of this section, to follow exactly what the authors are saying. Is there a simpler way to summarize this for the general (non-modeling expert) reader.

While we agree that this section is dry and strenuous for biologists we strongly believe that this level of detail should be provided as this modeling approach is the main novelty of this ms. Those readers interested in using this approach most likely will appreciate the provided detail. These details are already written in a non-modelling expert way. References to obtain additional details on the algorithms are provided, as well as the code for the applied models (see github).

Results:

Lines 337-342, and Figure 3: First I'm not sure the point of the paragraph, other than trying to explain (not very well), why the R2 value between BCC and Chl A was low? I also think Figure 3 is confusing...it necessary to be in manuscript? A rule of a good figure is that it should be easy to interpret the main takeaway, this is not true for Fig. 3. Add labels directly on Figure? Move to supplement?

We have removed the explanations about denoising and its effect on R-squared values, as recommended.

We agree Figure 3 can be confusing. In order to clarify the ideas conveyed by the figure, we have added a panel presenting what an expected linear relationship between beta diversity and the chlorophyll a gradient would look like, in contrast to what is actually observed. While the spatio-temporal approach takes individual ASVs as inputs as opposed to the model behind Figure 3 taking in a single BCC variable, the Bray-Curtis distance, the patterns of variation in both ASV distributions and beta diversity values are strongly non-linear along the gradients of interest in this manuscript. We believe it necessary to show why it would be so difficult to model accurately the observed

non-linear patterns using correlations and matrix inversion-based regressions and why tree-based methods are far better suited here, as was chosen for the spatio-temporal modeling approach. This is an essential figure on how to conceptualize classification systems and how ML techniques can be used to facilitate their modeling and interpretation.

We now mention the main results of Figure 3 in the results section and further explanation is added to the figure caption. We also refer to a recently published paper by our group on how these figures can be interpreted. Furthermore, more detail on how to interpret this figure is found in the discussion.

Discussion:

485-487: Expand here. Cite papers that have used similar modeling, explain how this study builds/is different. It is important to capitalize on novelty of study.

We have moved this important discussion to the beginning of the discussion and also expanded the perspective.

Lines 505-525: Very repetitive paragraph, revise.

We have removed the last two sentences which were highly repetitive.

Reviewer #3 (Remarks to the Author):

The manuscript's authors solved the issues I raised in the first revision and I consider it suitable for the publication.

We want to thank the reviewer for his valuable comments in the previous round of review.